# D3BT: Dynamic 3D Body Transformer for Body Fat Percentage Assessment

Yijiang Zheng
*Dept. of Computer Science*
*The George Washington University*
Washington, DC
yijiangzheng@gwu.edu

Zhuoxin Long
*Dept. of Statistics*
*The George Washington University*
Washington, DC
zlong66@gwu.edu

Boyuan Feng
*Dept. of Computer Science*
*The George Washington University*
Washington, DC
fby@gwu.edu

Ruting Cheng
*Dept. of Computer Science*
*The George Washington University*
Washington, DC
rcheng77@gwu.edu

Khashayar Vaziri
*Dept. of Surgery*
*The George Washington University*
*Medical Faculty Associates*
Washington, DC
kvaziri@mfa.gwu.edu

James K. Hahn
*Dept. of Computer Science*
*The George Washington University*
Washington, DC
hahn@gwu.edu

*Abstract*—3D body scan has been adopted for body composition assessment due to its ability to accurately capture body shape measurements. However, the complexity of mesh representation and the lack of fine-shape descriptors limit its applications for fat percentage analysis. Most studies rely on algorithms applied to anthropometric values derived from 3D scans, such as multiple girth measurements, which fail to account for the body's detailed shape. To address these issues, we explore the feasibility of using point cloud representation. However, few existing point-based methods are aimed at the human body or regression tasks.

In this study, we introduce a new model, D3BT, which utilizes a transformer-based network on the body point cloud to efficiently learn shape information for regional and global fat percentage regression tasks. The model dynamically divides the points into voxels for enhanced transformer training, providing higher density and better alignment across different subjects, which is more suitable for body shape learning. We evaluate different models predicting body fat percentage from 3D body scans, using ground truth data from dual-energy x-ray absorptiometry (DXA) reports. Compared to traditional methods that depend on anthropometric measurements and other point-based approaches, the proposed model shows superior results. In extensive experiments, the model reduces Root Mean Square Error (RMSE) by an average of 10.3% and achieves an average R-squared score of 0.86.

*Keywords— DXA, fat percentage, regression, point cloud, transformer learning, 3D body scan.*

## I. INTRODUCTION

High body fat percentage has been shown to be related to numerous diseases such as type 2 diabetes and hepatic steatosis [1][2]. There are various non-invasive techniques to evaluate the fat percentage in both clinical settings and home self-assessments. Medical imaging methods like magnetic resonance imaging (MRI), computed tomography (CT), and dual-energy x-ray absorptiometry (DXA) [3] can accurately calculate body composition. Bioelectrical impedance analysis (BIA) applies electric current for body composition calculation [4]. Anthropometric biomarkers like the waist-to-height ratio are strongly correlated with fat percentage [5]. However, despite their accuracy, medical imaging procedures are costly and impractical for telehealth applications. BIA and anthropometric measurements lack sensitivity due to the absence of standard prediction equations for calculating fat percentage [6][7]. The accuracy of BIA can fluctuate greatly due to factors like hydration levels and recent exercise, leading to inconsistent results, especially in individuals with obesity [8]. Consequently, there is a need for reliable and low-cost body composition assessment tools suitable for large-scale studies and digital health purposes [3].

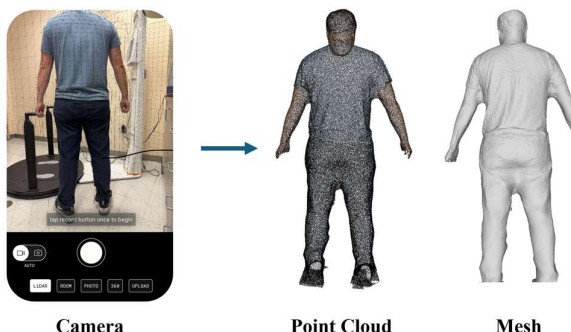

Fig. 1. An example of using a cell phone to capture body shape with the application Polycam, which provides 3D scanning capabilities utilizing both LiDAR and regular cameras. The outcome of the scan can be in either point cloud or mesh format.

One alternative tool for estimating body composition is the 3D body scan, which utilizes inexpensive equipment to accurately capture body shape [9][10]. These scans can measure body shape within minutes, with advanced devices achieving up to 97.5% accuracy in anthropometric measurements [11]. Accessible and affordable devices enable users to easily monitor the progress of body shape changes at home. The 3D body shape is less susceptible to short-term physiological changes, providing a more reliable option for detailed assessments. Fig. 1 shows the use of an iPhone (iPhone 15 Pro Max; Apple Inc) with the Polycam application (LiDAR & 3D Scanner; Polycam Inc) for body scanning using either LiDAR or regular cameras.

The utilization of 3D body scans to predict fat percentage has been investigated in several studies [12][13][14]. Most research on body composition using 3D body shapes applies machine learning algorithms to anthropometric values derived from the mesh representation of body scans, with the accuracy of results being highly dependent on the precision of the mesh [12][15]. However, generating a mesh representation requires

This work was supported in part by National Institutes of Diabetes and Digestive and Kidney Diseases (NIDDK) of the National Institutes of Health under grants R01DK129809.

additional processing steps compared to point clouds. A key challenge in utilizing 3D body meshes is the extraction of body features. Effective shape descriptors are important for accurate body composition estimation. Studies simplify the 3D information into one-dimensional features like volume and surface area [16], leading to the loss of rich details inherent in 3D meshes. Several studies also apply deep learning methods directly to 3D meshes [17][18]. However, meshes consist of vertex and edge information, requiring more computational resources (e.g., GPU memory size) to run the algorithms. Point clouds, in contrast, preserve the raw spatial information of the 3D scans and allow algorithms to be applied directly to them. It potentially offers a more straightforward and resource-efficient method for analyzing body fat percentage.

*A. Previous Work*

Many studies have employed 3D body scans to predict body composition using a variety of methods. For instance, a study by Bennett et al. indicates a strong agreement between fat mass values from DXA and those calculated from 3D body scans [19]. Pleuss et al. perform principal component analysis (PCA) and k-means clustering on shape anthropometry for body fat analysis [12]. Additionally, anthropometric features extracted from 3D body scans generated by smartphones are used to predict appendicular lean mass through the least absolute shrinkage and selection operator (LASSO) regression model [15]. Neural networks, based on anthropometric features, have also been applied to estimate appendicular skeletal muscle mass and liver fat percentage [20][21]. 3D body scans have been utilized across diverse populations as well. Tian et al. apply PCA on the meshes to parameterize the 3D shape space and utilize linear regression to predict fat percentage in pediatric populations [13]. Furthermore, 3D body scans can estimate body fat percentage in patients with conditions such as malnutrition and sarcopenia [22]. However, anthropometric features from 3D meshes and feature reduction methods like PCA can result in the loss of detailed shape information. In the neural network models mentioned, these extracted features are often treated as independent variables, neglecting the correlations among them. Moreover, there are insufficient studies focused on investigating fat percentages in specific regional body parts, which could provide more insights into fat distribution.

Using point clouds instead of features from the 3D mesh offers raw, detailed information about body shape. However, learning useful information from the points is challenging due to their unordered nature [23][24]. PointNet addresses this challenge by applying multi-layer perceptrons (MLP) and symmetry functions to the unstructured point set to achieve permutation invariance [23]. Building on this, PointNet++ employs local grouping and hierarchical structures to capture regional point information, followed by max-pooling to summarize these regions [24][25]. Transformers, which have had huge success in the natural language processing (NLP) domain, have been adapted for point cloud data. Proposed by Yan et al., PointASNL uses the self-attention mechanism for adaptive sampling within local point groups to fine-tune the distribution of points [26]. Point cloud transformer (PCT) replaces self-attention with an offset-attention structure, leveraging the Laplacian matrix, like graph convolution networks, to enhance feature learning [27]. The point transformer (PT-v1) combines self-attention and vector-attention, followed by local aggregation modules to facilitate information exchanges among points [28]. By splitting the vector-attention, group attention helps the efficiency and

generalization of the model [29]. Inspired by vision transformer (ViT) [30], Wu et al. serialize point clouds into a structured format and apply transformers to fixed-size patches [31].

However, these models were initially designed for object classification and segmentation tasks, which deal with more irregular inputs than human body scans. Regression tasks, such as predicting body composition, require models to capture more detailed spatial structures. Therefore, while point clouds offer a rich representation of spatial information, effectively analyzing point data for regression tasks like body fat percentage estimation remains complex. Further refinement of models is necessary for accurate body composition analysis.

To address the challenges in effectively utilizing point cloud data for fat percentage analysis, we propose a dynamic 3D body transformer network (D3BT). This network adopts varying voxel sizes to create denser patches for transformer learning, providing better spatial alignment for the human body compared to fixed voxel sizes. Each patch includes features derived from point local aggregation, and position embedding, obtained through the patch indices, resembling the structure of the ViT.

The main contributions of the paper are as follows:

1) We explore the feasibility of applying point cloud deep learning methods, moving beyond traditional circumference-based approaches, to estimate body fat percentages using 3D body scans.

2) We propose a novel architecture that utilizes dynamic voxel sizes to construct suitable patch groups, enhancing feature representation for transformer learning specifically for the human 3D body shape.

3) We demonstrate the efficiency of the proposed model for both local and global fat percentage regression calculations compared to existing models. In addition, we provide visual explanations of the model's performance by highlighting the critical regions that are most important in determining body fat percentage.

## II. METHOD

Our model addresses the challenges of applying transformer structures to human point clouds by partitioning the unstructured points into a fixed number of voxels after local aggregation training. This approach is inspired by advancements in the ViT, where images are divided into uniform patches. For example, images are split into nine patches, each containing 16*16 pixels, resulting in equal feature sizes after flattening. It allows for the easy infusion of position embedding features. This method takes advantage of the spatial relationships inherent in image data. However, irregular point clouds require additional considerations for patch creation. Two main issues arise naturally:

1) Point clouds do not conform to structured grid-like image pixels, making it difficult to partition them into fixed patches without creating sparse or inconsistent patches.

2) Maintaining a consistent feature size for each patch is not straightforward.

*A. Transformer*

Transformer and self-attention have been successfully

applied to point cloud learning in classification and segmentation tasks. In self-attention, given the input features $X \in \mathbb{R}^{N \times D_{in}}$, the query matrix $Q$, key matrix $K$, and value matrix $V$ can be calculated by $X$ and three learnable linear transformations: $W_Q$, $W_K$ and $W_V$ by:

$$(Q, K, V) = X \cdot (W_Q, W_K, W_V) \qquad (1)$$

where $W_Q, W_K \in \mathbb{R}^{D_{in} \times D_k}$, $W_V \in \mathbb{R}^{D_{in} \times D_{in}}$. $N$ is the input data size, $D_{in}$ is the input dimensions and $D_k$ is the dimension of the vectors in the $Q$ and $K$.

The self-attention weight is calculated as:

$$A_{weigh} = Softmax\left(\frac{Q \cdot K^T}{\sqrt{D_k}}\right) \qquad (2)$$

The self-attention output $O_a$ is calculated by the attention weight and the value matrix via dot-product:

$$O_a = A_{weight} \cdot V \qquad (3)$$

However, preparing well-structured input features $X$ from point clouds remains challenging. In ViT, images are split into $N$ patches, each containing the same number of pixels, resulting in a structured and consistent input for the transformer architecture. While this grid representation is straightforward for 2D image data, it becomes much more complex for 3D point clouds.

*B. Dynamic Voxel*

For the first problem, to partition the points into fixed patches like 2D grids in image data, we intuitively utilize voxelization to represent patches. Voxels can organize unordered spatial point information into structured voxel features. However, most studies apply a uniform voxel size (cube) for the input point cloud [32]. Moreover, for the transformer structure, fixed voxel numbers are important, similar to fixed sentence lengths in NLP and fixed patch numbers in ViT, as this consistency allows for the easy incorporation of position embedding information. Using the cube format to generate fixed voxel numbers can be approached by either deforming the input points or applying a larger voxel size to cover whole-body scans for each subject. Rescaling the point cloud to fit fixed-number voxels can dramatically alter the shape structure, as shown in Fig. 2(b). Unlike classification and segmentation tasks, where some deformation might be tolerable, changing the body shape can seriously impact the accuracy of regression tasks such as fat percentage estimation. Increasing the voxel size, on the contrary, despite maintaining the original shape, has lower density results, as shown in Fig. 2(c).

In Fig. 2(d), instead of the cube format, dynamic voxel sizes can maintain a consistent number of voxels across different body scans, providing a denser representation. The voxel size adjusts for each body scan, ensuring that the body shape remains undistorted. The voxel size for each axis can be calculated based on the range of the coordinates and the desired number of voxels along that axis. For the X-axis, the size of the voxel $V_x$ along the X-axis can be determined by:

$$V_x = (X_{max} - X_{min})/N_x \qquad (4)$$

where $X_{max}$ and $X_{min}$ are the maximum and minimum coordinates along the X-axis, respectively. $N_x$ is the pre-defined number of voxels along the X-axis. Similarly, the voxel sizes along the Y-axis ($V_y$) and Z-axis ($V_z$) can be calculated in the same way. Each subject will have different voxel sizes but maintain the same total number of voxels $N_V$ without deforming the body shape. Along with offering a denser voxel representation, another reason for opting for dynamic voxel sizes in human body scans is the enhanced alignment of body regions across different subjects. Given that subjects are typically scanned in a standing posture with lifted arms in 3D body scans, voxels sharing identical indices are more likely to correspond to the same body regions across different subjects. The details of the region alignment are provided in the experiment section.

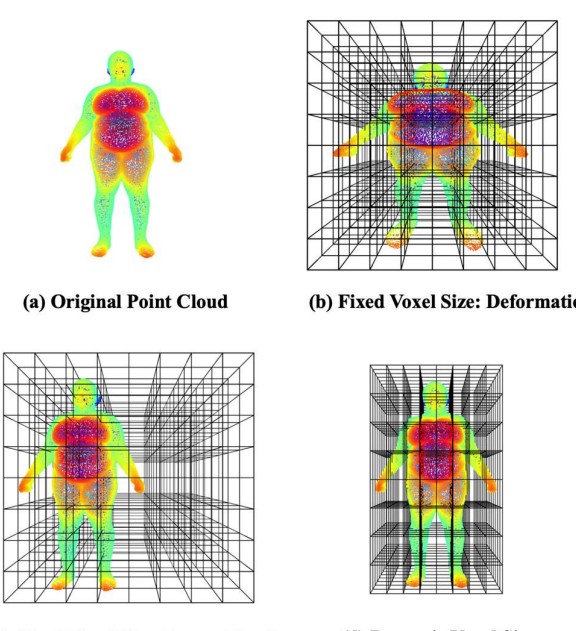

**(a) Original Point Cloud**   **(b) Fixed Voxel Size: Deformation**

**(c) Fixed Voxel Size: Larger Voxel**  **(d) Dynamic Voxel Size**

Fig. 2. An overview of different types of point cloud voxelization. All types have 8*8*8 voxel numbers. With fixed voxel sizes, subfigures (b) and (c) showcase point deformation and the use of a larger voxel size, respectively. Subfigure (d) illustrates the outcome of dynamic voxelization. The color of the points indicates the depth information.

*C. Dynamic 3D body Transformer Network*

In the ViT, patch division is applied directly to raw image pixels. However, in our experiment, we found that directly using point normal vectors as patch input features does not perform well. We suspect this is due to the spatial information difference between image grids and point voxelization. To address this, local aggregation layers are applied before voxelization to improve local feature representation. The structure of D3BT, as depicted in Fig. 3, incorporates local aggregation layers adopted from PointNet++ [24]. Input points undergo Farthest Point Sampling (FPS), followed by the aggregation and grouping of local neighbors' features via the Ball-Query algorithm. The resample ratios for the two aggregation layers are 0.5 and 0.25, respectively. After the local aggregation layers, the number of points $N_P$ decreases from 4096 to 512 while each point's feature size $C_P$ increases from 3 to 256. This procedure enables the model to retain essential geometric features of the body while significantly reducing computational complexity. Additionally, it functions as an efficient pooling process, further enhancing overall performance.

Maintaining consistent feature sizes across patches in images is achieved by flattening the pixels of each patch.

## Point Features Extraction

$N_P=4096, C_P=3$    $N_P=2048, C_P=128$    $N_P=512, C_P=256$

## Voxelization and Transformer

Position Embeddings

$N_V = N_x * N_y * N_z, C_V = 256$

Patch Features

Masks

| | | | | |
|---|---|---|---|---|
| ■ Local Aggregation | ■ Position Encoding | ▢ MLP | ■ Transformer | ■ Pooling |

FPS | Ball Query and Group | MLP

Self-Attention | MLP

Fig. 3. The structure of the dynamic 3D body transformer network. Initially, two layers of local aggregation are applied to resample the point cloud, enriching the points features. The point cloud is then dynamically split into fixed-number patches (voxels) using a pooling operator. Before transformer layers, position embeddings are added to each patch. Masks containing valid voxel indices guide the attention score calculation through all four transformer layers. MLP: multi-layer perceptions.

However, each voxel may not uniformly contain the same number of points. To address the second issue, we utilize operators such as mean or max pooling for each voxel, ensuring uniform voxel feature sizes $C_V$. These voxel features are then added with position embeddings of matching feature sizes. During our experiment, we observed that single-head attention generally outperforms multi-head attention. Unlike image data, where each patch has pixel values, some voxels might not contain any points. Hence, similar to NLP's padding token, we use the mask vectors to denote non-empty voxel indices. Consequently, during attention score calculation, empty voxels are disregarded.

## III. EXPERIMENT

In this section, we provide a detailed description of the dataset and experimental setup, including evaluation metrics and training hyperparameters. We also outline the approach for integrating demographic features into the models.

### A. Dataset

To assess the performance of our model, we conduct experiments on a dataset that includes both normal BMI subjects [33] and bariatric patients (BMI > 30). The bariatric patients are recruited from the George Washington University Hospital. Each subject has undergone a DXA (GE Lunar iDXA; GE Healthcare Lunar) scan to obtain their body fat percentage information. Additionally, 3D body scans of bariatric patients are obtained using a commercial scanner, Fit3D (ProScanner v6.0; Fit3D Inc), before their surgeries. As directed by the scanner, subjects are instructed to wear tight-fitting clothing and stand in an "A" posture during scanning. Instead of raising their arms to shoulder height, they slightly lift their arms to hold adjustable handles, keeping their arms straight and stationary throughout the scan for better alignment. The dataset consists of a total of 620 scans (486 normal BMI and 134 obese).

A DXA report example is presented in Fig. 4. Notably, for obese patients whose width exceeds the scanning bed limits, only the right side of the body is scanned [33]. The left side's body fat percentage values are estimated symmetrically. DXA reports provide body fat percentage information for five

different local regions as well as the total body fat percentage. These reports serve as the ground truth for the experiments. The study was approved by the Institutional Review Board (IRB) of The George Washington University.

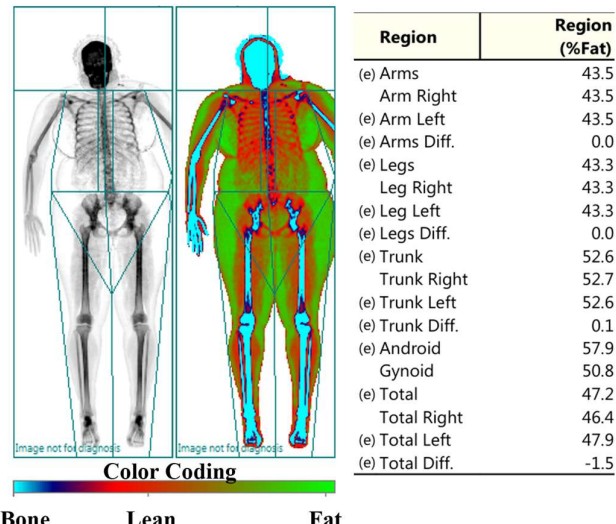

Color Coding

Bone        Lean        Fat

| Region | Region (%Fat) |
|---|---|
| (e) Arms | 43.5 |
| Arm Right | 43.5 |
| (e) Arm Left | 43.5 |
| (e) Arms Diff. | 0.0 |
| (e) Legs | 43.3 |
| Leg Right | 43.3 |
| (e) Leg Left | 43.3 |
| (e) Legs Diff. | 0.0 |
| (e) Trunk | 52.6 |
| Trunk Right | 52.7 |
| (e) Trunk Left | 52.6 |
| (e) Trunk Diff. | 0.1 |
| (e) Android | 57.9 |
| Gynoid | 50.8 |
| (e) Total | 47.2 |
| Total Right | 46.4 |
| (e) Total Left | 47.9 |
| (e) Total Diff. | -1.5 |

Fig. 4. An example of a DXA report from an obese subject. The report provides fat percentages in various regions. (e): estimated.

For analysis purposes, the 3D body scans are segmented into five anatomical parts: left arm, right arm, left leg, right leg, and trunk, corresponding to the DXA regional splitting. We uniformly sample 4096 points for each body part with each point characterized by its coordinates and vertex normal vector information. In addition, circumference features are extracted, encompassing 64 levels for each region to represent anthropometric measurements [20]. We combine the features from all five regions as whole-body features.

### B. Voxelization Comparison

To evaluate the differences between the three voxelization approaches mentioned above, we conduct two tests: density and alignment. The voxel number is configured to 8*16*4,

consistent with the training network. In the density test, we compare the ratio of non-empty voxels number to the total number of voxels. For the alignment test, we randomly select 100 pairs of 3D body scans, with each pair including two different subjects. We use the mean Intersection over Union (mIoU) score to assess the overlap results in the five segmented parts. For example, we first calculate the voxel indices for the trunk region of two different subjects and then determine the IoU value for these two sets of voxel indices. This process is repeated for the other four local regions, and the results are averaged.

### C. Training Network

For the fat percentage regression tasks, we include several comparison models. Non-point models consist of a model with only demographic features (DF) as inputs and a model with level circumferences (LC). To ensure a fair comparison with other point-based models, instead of traditional regression models, these two features are trained using deep learning architectures consisting of three linear layers. We also compare our results with popular point cloud deep learning models: PointNet (PN), PointNet++ (PN++), and Dynamic Graph CNN (DGCNN). In addition, we evaluate transformer-based point cloud models: Point Cloud Transformer (PCT), Point Transformer version 1 (PTv1), and Point Transformer version 2 (PTv2). Since the point cloud is voxelized, the model 3DCNN [35] is inherently included. In medical contexts, it is common to apply transfer learning on smaller datasets [36][37]. For transfer learning models, we pretrain the PN++ model on the ModelNet40 dataset [35], which includes 12,311 3D objects from 40 categories. We compare two types of transfer learning, one by simply combining the learned point features with MLP (TL) and the other by applying our voxelization and transformers on the learned features (TL + Ours).

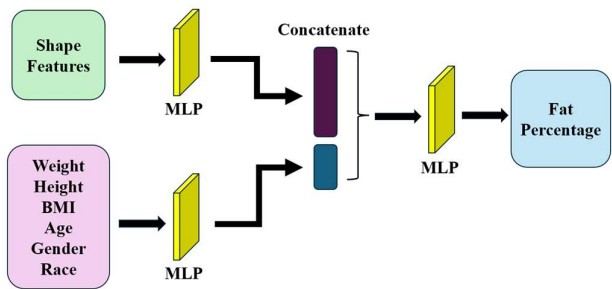

Fig. 5. The pipeline for integrating shape features with demographic information in models. The two sets of features are combined using standard concatenation, followed by additional training with linear layers for regression estimation.

Since body fat percentage is influenced by demographic information [20], particularly gender, we incorporate demographic values: age, weight, height, gender, and race into all training models except the DF model through linear layers. Incorporating demographic factors helps capture variations in body composition that are often associated with these variables. These features are easily accessible and provide valuable context, allowing models to deliver more accurate predictions and converge more quickly, as they don't need to infer these factors solely from body shape. Including BMI as a feature enhances prediction performance by capturing its direct relationship to fat percentage and introducing non-linear patterns. To even feature distribution, race is categorized into three groups: white, black, and others. Fig. 5 demonstrates the

integration of shape features with demographic features. The combined features are then fed into the final MLP for fat percentage estimation.

### D. Training Setup and Hyperparameters

We evaluate the performance of all models with Root Mean Square Error (RMSE) and R-squared score. In the experiment, we apply 5-fold cross-validation, testing after each epoch and selecting the best epoch result [23][24]. We choose the Mean Square Loss function and the Adam optimizer with an initial learning rate of 0.001. The learning rate decreases every 10 epochs with a decay factor of 0.7. The batch size is 16, and the number of epochs is 100. The voxel number is set to 8*16*4, and we report the impact of different voxel numbers as part of our ablation studies. All training is performed using Pytorch, with all layers implemented and run on one Nvidia RTX-4090 GPU.

## IV. RESULTS AND DISCUSSION

### A. Density and Alignment

Table I compares the differences between the three voxelization approaches in terms of voxel density and alignment. Dynamic voxelization not only achieves a higher density ratio but also maintains better alignment between different subjects. The deformation method preserves density but performs poorly in alignment, whereas the larger-voxel approach maintains alignment but sacrifices density.

TABLE I. VOXEL DENSITY RATIO AND MEAN IoU RESULTS OF THREE DIFFERENT TYPES OF VOXELIZATION APPROACHES (VOXEL NUMBER: 8*16*4).

|  | Density Ratio | Mean IoU |
| --- | --- | --- |
| Deformation | 40.052% | 0.195 |
| Lager Voxel | 15.796% | 0.477 |
| Dynamic | **42.032%** | **0.622** |

### B. Fat Percentage Prediction

The experimental results for fat percentage assessment from different models are presented in Table II. Among the existing point-based models, PointNet++ and 3DCNN demonstrate superior performance across all regions, surpassing the network that relies solely on demographic features by a large margin. Notably, the circumference-based network achieves relatively high accuracy compared to existing point-based models. Our model delivers the most accurate results in all body parts except for the left arm region. Specifically, it achieves the lowest RMSE in whole body fat percentage at 4.22%, with an R-squared value of 0.89. It reduces the RMSE by an average of 10.3% compared to model LC. Transfer learning does not help reduce error rates, which may be attributed to the discrepancy in the domain of the pretrained dataset and the specific task of the experiment, as the regression task requires capturing more details of shape information. Circumference and point-based models outperform the demographic-only network highlights the importance of detailed shape information in accurately estimating fat percentage. Relying solely on demographic data, such as age, height, and weight, without considering physical shape details, is insufficient to capture body shape variations needed for precise predictions.

TABLE II.    COMPARISON WITH EXISTING MODELS ON WHOLE BODY AND FIVE REGIONAL BODY PARTS IN FAT PERCENTAGE ESTIMATION. ALL RESULTS ARE FROM 5-FOLD CROSS-VALIDATION. RMSE (FAT PERCENTAGE %) AND R-SQUARED SCORE ARE SELECTED AS EVALUATION METRICS.

| Model | Whole | | Left Arm | | Right Arm | | Left Leg | | Right Leg | | Trunk | |
|---|---|---|---|---|---|---|---|---|---|---|---|---|
| | RMSE | $R^2$ | RMSE | $R^2$ | RMSE | $R^2$ | RMSE | $R^2$ | RMSE | $R^2$ | RMSE | $R^2$ |
| DF | 5.399 | 0.820 | 6.046 | 0.768 | 5.771 | 0.787 | 5.679 | 0.772 | 5.624 | 0.780 | 6.572 | 0.809 |
| LC | 4.305 | 0.885 | **4.925** | **0.846** | 5.090 | 0.834 | 4.746 | 0.841 | 6.594 | 0.697 | 6.355 | 0.822 |
| PN | 5.267 | 0.829 | 5.804 | 0.786 | 5.654 | 0.795 | 5.409 | 0.793 | 5.599 | 0.782 | 6.417 | 0.818 |
| PN++ | 4.545 | 0.872 | 5.249 | 0.825 | 5.230 | 0.825 | 4.901 | 0.830 | 5.007 | 0.825 | 5.423 | 0.870 |
| DGCNN | 4.998 | 0.846 | 5.672 | 0.795 | 5.586 | 0.800 | 5.378 | 0.795 | 5.373 | 0.799 | 6.082 | 0.837 |
| 3DCNN | 4.574 | 0.871 | 5.401 | 0.814 | 5.251 | 0.823 | 4.719 | 0.843 | 4.837 | 0.837 | 5.515 | 0.866 |
| TL | 4.772 | 0.859 | 5.714 | 0.792 | 5.492 | 0.807 | 5.019 | 0.822 | 4.971 | 0.828 | 5.856 | 0.848 |
| PCT | 4.939 | 0.849 | 5.697 | 0.794 | 5.724 | 0.790 | 5.328 | 0.799 | 5.326 | 0.802 | 6.339 | 0.822 |
| PTv1 | 5.041 | 0.843 | 5.763 | 0.789 | 5.423 | 0.812 | 5.397 | 0.794 | 5.459 | 0.792 | 6.235 | 0.828 |
| PTv2 | 4.757 | 0.860 | 5.645 | 0.797 | 5.628 | 0.797 | 5.536 | 0.783 | 5.436 | 0.794 | 6.132 | 0.834 |
| TL + Ours | 4.589 | 0.870 | 5.251 | 0.822 | 5.159 | 0.829 | 4.720 | 0.842 | 4.862 | 0.835 | 5.382 | 0.872 |
| Ours | **4.223** | **0.890** | 5.195 | 0.828 | **4.969** | **0.842** | **4.569** | **0.852** | **4.496** | **0.859** | **5.262** | **0.878** |

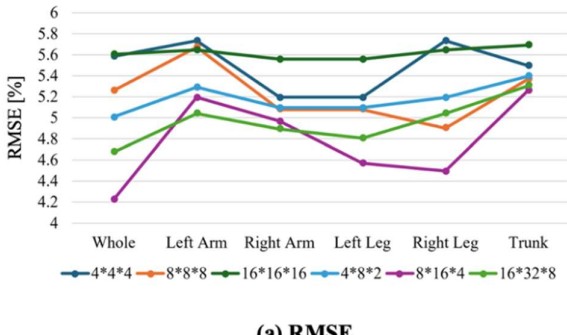

**(a) RMSE**

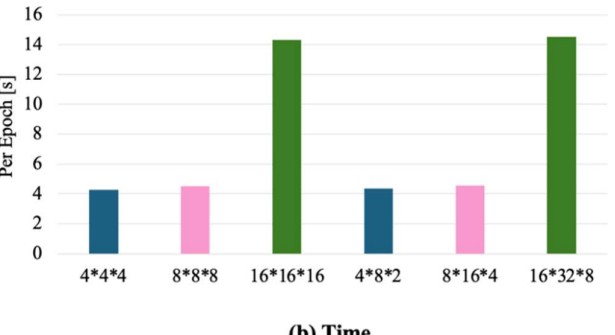

**(b) Time**

Fig. 6.    The RMSE results and training times under different voxel number configurations. Subfigure (a) shows the RMSE result for six body parts. Subfigure (b) presents the time consumed per epoch, with models having the same total voxel numbers indicated in the same color.

### C. Ablation Experiment

#### 1) Voxel Numbers

To test the impact of voxel numbers on training results, we compare the accuracy and training time across different voxel number settings along three axes. Training time is measured in seconds for one epoch. Fig. 6 presents the results. The network with voxel numbers set to 8*16*4 demonstrates the best overall performance. Models with equal voxel numbers for each axis generally perform worse than those with the same total voxel numbers but different distributions across each axis. This is likely due to the human standing posture, where height > width > thickness, making the 8:16:4 ratio more effective for evenly splitting the points into voxels. With the same total voxel numbers, training times are similar despite variations in the three axes. Increasing the total voxel numbers can significantly impact training time without necessarily guaranteeing better results.

#### 2) Network Structure

To prove the necessity of the local aggregation modules, we design a network (V3T) that applies voxelization at the beginning. In this network, the normal vector features are directly integrated into the patch features, like the ViT structure, where raw pixel features are forwarded into the patch features. Additionally, we design another network (LT) in which the voxelization and transformers are applied immediately after each local aggregation layer. The LT network has two local aggregations, two voxelizations, and two transformer modules. The first transformer voxel number is 8*16*4, while the second is 4*8*2 for larger receptive fields. Table III presents the average RMSE over local regions and the overall body fat percentage for the three structures. The results clearly show that the V3T model, which lacks local aggregation modules, performs worse than our model. This may be due to the inadequacy of the normal vector for transformer learning. With local aggregation operations, each point can integrate its neighbor region information to enrich feature presentation. Additionally, the LT model, with its two

transformer modules, increases the RMSE, suggesting that adding transformers sequentially improves results compared to embedding transformers within local aggregation layers.

TABLE III. COMPARISON OF THREE NETWORK STRUCTURES.

|  | Mean RMSE |
| --- | --- |
| V3T | 5.118 |
| LT | 5.502 |
| Our | **4.786** |

### D. Visualization

We analyze the attention representations from the first layer [31] as part of understanding how the transformer processes 3D body scan data for whole-body fat percentage predictions. Fig. 7 visualizes the normalized attention scores of the voxels across three view angles for each individual scan. The visualization reveals that the head and feet regions receive lower attention scores compared to the central areas, indicating they may be less significant in predicting fat percentage. This aligns with typical fat distribution patterns, as these regions generally contain less fat. Additionally, the model assigns similar importance to voxels from both the front and back of the body. Notably, there are no significant differences in the attention map distribution between obese and normal BMI populations.

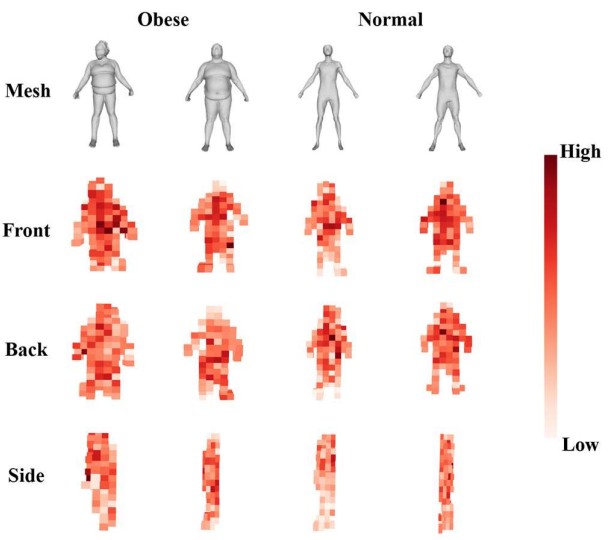

Fig. 7. Representative examples of attention scores on the voxels from the first transformer layer. The figure includes three view angles: front, back, and side. The first two columns display voxels from the obese population, while the remaining columns represent individuals with normal BMI. Empty voxels, which are ignored during the transformer calculation, are not shown.

## V. CONCLUSION

In this paper, we explore various point cloud deep learning models for body fat percentage assessment in regression tasks. We propose the model D3BT, a transformer-based neural network that dynamically partitions the body point cloud into fixed-number voxels to reduce fat percentage estimation error. In detail, the dynamic voxelization design enhances voxel density and region alignment, which are important for the position embedding in the transformer structure. The architecture improves the sensitivity of the network by optimizing the voxel number ratio along three axes to suit the human standing posture. Through extensive experiments, the algorithm demonstrates its effectiveness in both local and global fat percentage estimation using real-world 3D body scans. The superiority of D3BT over existing models suggests the potential for applying point cloud techniques to calculate fat percentage in real body scans without the need for clinical visits. This could be integrated into digital health routines, allowing individuals to monitor their body composition changes at home with affordable and accessible devices.

Despite the promising results, this study has some limitations. One key limitation is the exclusion of certain populations, such as the geriatric population, whose body composition is often significantly impacted by sarcopenia. Expanding the dataset is essential to improve the model's generalizability. Another limitation is the potential variability introduced by different scanning devices, as they may require specific postures or produce 3D images at varying resolutions, potentially affecting the outcomes. Future work, particularly in telehealth applications, should involve testing the model with 3D body scans obtained from smartphones. Validating these results against commonly used methods like BIA will be important to establish the model's reliability. As 3D body scans become more prevalent, future research should also investigate point-based models to assess other body compositions, such as lean mass.

### ACKNOWLEDGMENT

The work is supported by the National Institutes of Health under grant number R01DK129809. The content is solely the responsibility of the authors and does not necessarily represent the official views of the National Institutes of Health.

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
