# OpenReview forum: "D3BT: Dynamic 3D Body Transformer for Body Fat Percentage Assessment"
_IEEE.org/EMBS/BHI/2024/Conference — IEEE BHI'24_

### Official Review · Reviewer_kPWH · 2024-08-08
**This paper is well aligned with the topics of the conference. It is well written and presents a work with clear potential application in future telemedicine solutions.**

**Overall Rating:** 7
**Confidence:** 3

**Other Quality Metrics:**

(a) Clarity of writing: good.
(b) Clinical Significance: good.
(c) Methodological Novelty: good.
(d) Experiments and Results: good.

**Questions For The Authors:**

Although there is no much space left, the paper could benefit from adding a discussion section or expanding the conclusions one (e.g.,maybe some of the figures could be reduced, eliminated or presented in some other way). This section could address deeper the results obtained and expand the information about the the limitations of the work done and future applications and work.

**Strengths:**

The work presents a new method for estimating the body composition using 3D body scans and point cloud representations. The work explores the way towards a future reliable and low cost method for measuring body composition.
The paper provides a clear analysis of the existing challenges, the proposed method and contributions of the work done, together with the obtained results, which are claimed to be good and promising.

**Summary Of The Paper:**

The paper describes an exploration of novel ways for estimating body fat percentages using point cloud deep learning methods and dynamic voxel sizes. It proposes a new method and demonstrates its efficiency compared to existing ones.

**Weaknesses:**

The paper could benefit from adding a discussion section or providing a more extensive discussion of the results as well as the limitations and future work.

---

> ### Author Rebuttal · Authors · 2024-09-01
>
> We appreciate the time and effort you invested in reviewing our manuscript and for your constructive feedback. Below, we address the points raised in your review:
>
> 1. Expanding discussion and conclusion
>
> We acknowledge the importance of a detailed discussion and conclusion and agree that expanding on the implications of our findings would enhance the paper. We will add a more extensive discussion for the experiment results, including comparisons of outcomes based on different input features. In addition, we will provide more information about the limitations of the study and potential challenges in real-world applications including areas where further research is needed for applying 3D scans on body composition assessment.
>
> 2. Figures adjustment
>
> To accommodate a more in-depth discussion of the results, we will adjust some figures in consideration of the paper's length restrictions.

---

### Official Review · Reviewer_5C9U · 2024-08-09
**This paper is a good contribution to the local body fat percentage estimation. The method is novel in the sense that it utilizes directly 3D body scan point cloud for a regression task. Comparative analysis indicates that the proposed method outperforms existing methods.**

**Overall Rating:** 8
**Confidence:** 3

**Other Quality Metrics:**

(a) Clarity of writing: excellent
(b) Clinical Significance: good
(c) Methodological Novelty: excellent
(d) Experiments and Results: excellent

**Questions For The Authors:**

The paper mentioned that for the body scan the subjects typically take the "A" posture, whereas for the data collection of this study the subjects are instructed to take the posture with their arms down. Could you please explain the reason for the posture choice, since it seems that the "A" posture is able to help more on the alignment of the body parts across various subjects?

**Strengths:**

1. The proposed method uses directly the point cloud data which avoids the resource-consuming conversion to and storage of the 3D mesh data.
2. The adoption of the dynamic voxelization facilitates the generalization of the algorithm to subjects with various physical shapes.
3. The accuracy of the proposed method is superior compared to related works for the majority of the body parts.

**Summary Of The Paper:**

This paper proposes a transformer-based approach that directly utilizes 3D body scan point cloud to the estimation of local and global body fat percentage.

**Weaknesses:**

Since the paper adopts the local aggregation of point cloud data after voxelization, it is a bit confusing about the difference in the data quality between the locally aggregated point cloud data and the 3D mesh data. Could you please explain the effects of the local aggregation on the precision of the point cloud data?

---

> ### Author Rebuttal · Authors · 2024-09-01
>
> We are grateful for your thorough review of our manuscript and the positive feedback on the strengths of our work. Below, we address the specific questions and concerns you raised:
>
> 1. Local aggregation
>
> The local aggregation of point cloud data after voxelization is a crucial step in our method. This approach allows us to retain the essential geometric features of the body while significantly reducing computational complexity. Although the aggregated point cloud data may not offer the same level of precision as 3D mesh data, it still captures the critical structural information needed for accurate analysis. While 3D mesh data provides additional edge details, the high demands on computing resources—both in terms of time and memory—limit its practical scalability. Moreover, point clouds facilitate more efficient pooling processes compared to mesh data, making them more suitable for our application. We will clarify this in the final manuscript to ensure the rationale behind our approach is well understood.
>
> 2. “A” posture
>
> We apologize for any confusion regarding the “A” posture. In our data collection for bariatric patients, we used the commercial 3D scanner Fit3D. In the scanning process, subjects slightly lift their arms to hold onto adjustable handles, rather than raising their arms to shoulder height. The handles are positioned to ensure the arms remain stationary and provide stability during scanning, which helps achieve better alignment. We will include a description of this posture in the final paper. For further details, the Fit3D website provides a demonstration of the standing posture used with their machine: Fit3D 3D Body Scanner (https://www.fit3d.com/3d-body-scanner).

---

### Official Review · Reviewer_cqps · 2024-08-10
**D3BT: Dynamic 3D Body Transformer for Body Fat Percentage Assessment**

**Overall Rating:** 7
**Confidence:** 3

**Other Quality Metrics:**

Clarity of Writing: Great
Clinical Significance: Fair
Methodological Novelty: Good
Experiments and Results: Great

**Questions For The Authors:**

- Are information like "race" necessary for the estimation, also BMI should be unnecessary, as height and weight are already inputs?
- Would it be possible to omit gender as input, and let the network estimate purely based on the body shape?
- Could you include BIA results for comparison in a few cases?

**Strengths:**

- Easy-to-follow method description
- Trained using a reliable source of data
- Method has the potentail to be used at home with high accuracy
- Promising results

**Summary Of The Paper:**

The authors developed a method to estimate the body fat percentage from demographic information and a 3D scan using MLPs.
Using a voxel representation calculated from the point cloud, they also can assign fat percentages to some body parts.
Their method yields high correlation and a low RMSE when compared to other state of the art methods.

**Weaknesses:**

- What is the benefit compared to BIA for home usage (accuracy?)

---

> ### Author Rebuttal · Authors · 2024-09-01
>
> Thank you for your thoughtful review of our manuscript and for acknowledging the strengths of our work. We appreciate your insights and the opportunity to address your questions and concerns.
>
> 1. Necessity of Demographic Inputs: “Race” and “BMI”
>
> The inclusion of demographic factors such as race and BMI was designed to capture variations in body composition that often correlate with these factors. Race and ethnicity can influence body fat distribution. Although height and weight can be used to calculate BMI, explicitly including BMI as a feature enhances prediction performance because it directly relates to fat percentage and provides non-linear relationships that help improve accuracy. We will clarify the rationale for selecting these demographic features in the revised manuscript.
>
> 2. Omitting Gender
>
> While it is technically feasible to omit gender as an input and rely solely on body shape for estimation, demographic features like gender and age are easily accessible and offer valuable context for the model. Including this information allows for more accurate predictions and faster model convergence, as the model doesn’t need to infer these factors from body shape alone. We will add this clarification to the final manuscript.
>
> 3. Comparison with BIA for Home Use (Accuracy) and Inclusion of BIA Results for Comparison
>
> We understand that Bioelectrical Impedance Analysis (BIA) is a widely used tool for estimating body fat at home. BIA's accuracy can fluctuate significantly based on factors like hydration levels, recent exercise, and food intake, leading to inconsistent results. Additionally, most BIA devices provide only a whole-body composition assessment, potentially overlooking variations in body fat distribution across different regions. In contrast, our method, which utilizes 3D body shape data, offers local body composition insights and is less susceptible to short-term physiological changes, making it a more reliable option for detailed analysis. We will include these points in the final manuscript.
> DXA is widely considered the "gold standard" for body composition analysis due to its high accuracy and detailed assessment of body fat percentage and distribution. Given the limitations in accuracy associated with BIA, we chose not to collect BIA data during this study. We plan to explore the inclusion of BIA information in future work.

---

### Decision · Program_Chairs · 2024-09-23

Accept